# Procalcitonin, C-reactive protein, neutrophil gelatinase-associated lipocalin, resistin and the APTT waveform for the early diagnosis of serious bacterial infection and prediction of outcome in critically ill children

Maryke J. Nielsen[1,2,3]*, Paul Baines[4,5], Rebecca Jennings[6], Sarah Siner[4], Ruwanthi Kolamunnage-Dona[7], Paul Newland[8], Matthew Peak[6], Christine Chesters[9], Graham Jeffers[7], Colin Downey[10], Caroline Broughton[1], Lynsey McColl[11], Jennifer Preston[7], Anthony McKeever[12], Stephane Paulus[1], Nigel Cunliffe[1,3,13], Enitan D. Carrol[1,3,14]

1 Institute of Infection, Veterinary and Ecological Sciences, University of Liverpool, Liverpool, United Kingdom, 2 Malawi-Liverpool-Wellcome Trust Clinical Research Facility, Blantyre, Malawi, 3 Liverpool Health Partners, Liverpool, United Kingdom, 4 Critical Care, Alder Hey Children's NHS Foundation Trust, Liverpool, United Kingdom, 5 Medicine, Ethics, Society & History, University of Birmingham, Birmingham, United Kingdom, 6 Clinical Research Business Unit, Alder Hey Children's NHS Foundation Trust, Liverpool, United Kingdom, 7 Institute of Life Course & Medical Sciences, University of Liverpool, Liverpool, United Kingdom, 8 Department of Pathology, Sidra Medicine, Doha, Qatar, 9 Clinical Biochemistry, Alder Hey Children's NHS Foundation Trust, Liverpool, United Kingdom, 10 Haematology, Royal Liverpool and Broadgreen University Hospital Trust, Liverpool, United Kingdom, 11 Select Statistical Services, Exeter, United Kingdom, 12 School of Medicine, University of Liverpool, Liverpool, United Kingdom, 13 Clinical Microbiology, Alder Hey Children's NHS Foundation Trust, Liverpool, United Kingdom, 14 Department of Infectious Diseases, Alder Hey Children's NHS Foundation Trust, Liverpool, United Kingdom

* m.nielsen@liverpool.ac.uk

## Abstract

### Objective

Bacterial Infections remains a leading cause of death in the Paediatric Intensive Care Unit (PICU). In this era of rising antimicrobial resistance, new tools are needed to guide antimicrobial use. The aim of this study was to investigate the accuracy of procalcitonin (PCT), neutrophil gelatinase-associated lipocalin (NGAL), resistin, activated partial thromboplastin time (aPTT) waveform and C-reactive protein (CRP) for the diagnosis of serious bacterial infection (SBI) in children on admission to PICU and their use as prognostic indicators.

### Setting

A regional PICU in the United Kingdom.

### Patients

Consecutive PICU admissions between October 2010 and June 2012.

**Data Availability Statement:** The data that support the findings of this study are available on request

from the University of Liverpool, Institute of Infection, Veterinary and Ecological Sciences Head of Operations iveshoo@liverpool.ac.uk The data are not publicly available due to restrictions imposed by collection of patient data from a named hospital, and therefore contains information that could compromise the privacy of research participants.

**Funding:** The study was funded jointly by the NIHR Liverpool Biomedical Research Centre in Microbial Diseases and the Alder Hey Charity awarded to EC. MJN is supported by a Wellcome Trust Research Training Fellowship (award reference 203919/Z/16/Z). The funders had no role in the study design, data collection and analysis, decision to publics, or preparation of the manuscript.

**Competing interests:** The authors have read the journal's policy and have the following potential competing interest: LMcC is a paid employee of Select Statistics. This does not alter our adherence to PLOS ONE policies on sharing data and materials. There are no patents, products in development or marketed products associated with this research to declare. The other authors declare that they have no competing interests.

**Abbreviations:** aPTT, Activated Partial Thromboplastin Time; AUC, Area Under the Curve; CI, Confidence Interval; CRP, C-Reactive Protein; ICU, Intensive Care Unit; IPPV, Invasive positive pressure ventilatio; LR, Likelihood Rati; NGAL, Neutrophil Gelatinase Associated Lipocali; NPV, Negative Predictive Valu; NRI, Net Reclassification Improvemen; OR, Odds Rati; PCT, Procalcitoni; PICU, Paediatric Intensive Care Uni; PELOD, Paediatric Logistic Organ Dysfunction Scor; PPV, Positive Predictive Valu; ROC, Receiver Operator Curve; SBI, Serious Bacterial Infectio; SIRS, Systemic Inflammatory Response Syndrom.

## Measurements

Blood samples were collected daily for biomarker measurement. The primary outcome measure was performance of study biomarkers for diagnosis of SBI on admission to PICU based on clinical, radiological and microbiological criteria. Secondary outcomes included durations of PICU stay and invasive ventilation and 28-day mortality. Patients were followed up to day 28 post-admission.

## Main results

A total of 657 patients were included in the study. 92 patients (14%) fulfilled criteria for SBI. 28-day mortality was 2.6% (17/657), but 8.7% (8/92) for patients with SBI. The combination of PCT, resistin, plasma NGAL and CRP resulted in the greatest net reclassification improvement compared to CRP alone (0.69, p<0.005) with 10.5% reduction in correct classification of patients with SBI (p 0.52) but a 78% improvement in correct classification of patients without events (p <0.005). A statistical model of prolonged duration of PICU stay found log-transformed maximum values of biomarkers performed better than first recorded biomarkers. The final model included maximum values of CRP, plasma NGAL, lymphocyte and platelet count (AUC 79%, 95% CI 73.7% to 84.2%). Longitudinal profiles of biomarkers showed PCT levels to decrease most rapidly following admission SBI.

## Conclusion

Combinations of biomarkers, including PCT, may improve accurate and timely identification of SBI on admission to PICU.

## Introduction

Invasive bacterial infections account for over a quarter of all deaths in PICU [1] whilst up to 31% of paediatric sepsis survivors are affected by disability at discharge [2]. Early recognition of sepsis and prompt anti-microbial therapy reduce mortality and duration of organ dysfunction [3–5], but indiscriminate antimicrobial use contributes to resistance [6, 7]. Differentiation of infective and non-infective causes of the systemic inflammatory response syndrome (SIRS) is an ongoing challenge for clinicians. A reliable marker, or combination of markers, that change early in bacterial infection, correlate with real-time clinical progression and have a rapid laboratory turn-around time is an urgent unmet clinical need.

Procalcitonin (PCT) has been shown in comparatively small studies to be a better diagnostic marker of bacterial infection in PICU than C-reactive protein (CRP) [8, 9] and to be a prognostic marker in meningococcal disease [10]. The biphasic activated Partial Thromboplastin Time (aPTT) waveform is the optical profile generated from changes in light transmittance during clot formation. It has been found in adults to be a more useful marker of sepsis than CRP alone, correlating with increasing risk of clinical deterioration and disseminated intravascular coagulation (DIC) [11–13] and to be abnormal in children with meningococcal sepsis [14]. Neutrophil gelatinase- associated lipocalin (NGAL), measured in plasma or urine, is a marker of acute kidney injury but also a promising marker of sepsis and multi-organ dysfunction in adults [15, 16] and neonates [17]. Resistin an adipokine which contributes to inflammation-induced insulin resistance, has been shown to correlate with sepsis severity in adults [18,

19]. The performance of the aPTT waveform, NGAL and resistin in paediatric infection is unknown.

## Objectives

The primary aim of this study was to determine the discriminative ability of PCT, aPTT waveform, NGAL and resistin individually, in combination and compared to CRP to diagnose serious bacterial infection (SBI) in children on admission to PICU. The secondary objectives were to investigate which variables, including admission and longitudinal biomarkers, are predictive of duration of ventilation, duration of and prolonged PICU stay, and 28-day mortality.

## Methods

### Study design and setting

A prospective, observational study was conducted on the PICU at Alder Hey Children's NHS Foundation Trust, Liverpool, U.K between October 2010 and June 2012.

### Participants

All patients from birth to 16 years admitted to the PICU were eligible for inclusion. Pre-term infants < 37 weeks corrected gestation, children predicted not to survive at least 28 days due to a pre-existing condition or with an existing directive to withhold life-sustaining treatment, children with end stage renal disease requiring chronic dialysis, end-stage liver disease, children admitted moribund and not expected to survive more than 24 hours and non-intubated elective admissions with a predicted duration of stay less than 24 hours were excluded.

Some participants in our study were also recruited to a concurrent multicentre randomised control trial, Control of Hyperglycaemia in Children in Paediatric Intensive Care (CHiP) [20].

### Ethical approval and sample size calculation

Written informed consent for participation was obtained from parents or guardians. Patients were followed up until day 28 after admission. The study was approved by University of Liverpool research ethics committee (REC reference number: 10/H1014/52).

Prior to study commencement data from the Paediatric Intensive Care Audit Network (PICANET) found sepsis in 15% of admissions. The pre-study power calculation was 1640 patients with 246 SBI events to achieve 80% sensitivity and specificity with 15% SBI incidence in an ROC curve analysis at 5% significance level (alpha = 0.05, one-sided).

### Data collection and outcomes measured

A case was defined as a child in whom an SBI was present on admission to PICU. SBI was defined in accordance with previous studies as a suspected or proven infection caused by any bacterial pathogen, or a clinical syndrome associated with a high probability of infection which included positive findings on examination, imaging or hospital laboratory tests [21, 22] (S1 Table in S1 File). Cases were discussed at a twice weekly PICU Infection multi-disciplinary team meeting which included the duty Microbiology, PICU and Paediatric Infectious Disease Consultants to determine the presence and timing of SBI. A paediatric infectious disease Consultant or research fellow reviewed cases in which there was disagreement amongst the team. Patients were followed up until day 28 to ensure misclassification had not occurred.

Sepsis and SIRS were defined in accordance with accepted international definitions [22]. New consensus definitions in adults have yet to be adopted in paediatrics [23]. Secondary outcome measures included 28-day mortality, duration of invasive positive pressure ventilation

(IPPV), duration of PICU stay and prolonged ICU stay. A participant was defined as having a prolonged ICU stay if their duration of stay was greater than or equal to the median. Paediatric logistic organ dysfunction score (PELOD), a quantitative score of organ dysfunction, was chosen as a surrogate marker for disease severity [24].

## Measurement of biomarkers

Four study biomarkers, comprising PCT, plasma NGAL, resistin and aPTT biphasic wave form (slope and light transmittance level at 18s) were measured daily for seven days following PICU admission from blood samples collected at the same time as routine investigations. CRP was measured on an Abbott Architect analyser in the hospital's clinical chemistry laboratory. Urine NGAL was measured in catheterised patients. Whole blood collected in lithium heparin and sodium citrate (for APTT waveform) microtubes was spun down at 3000 rpm for 10 minutes within 2 hours of collection, and the plasma was stored in aliquots for biomarker determination at the end of the study, when the PCT NGAL, resistin analysis was performed. All samples were stored at -80˚C until analysis. CRP is currently the routine sepsis biomarker used in the U.K. and as such results were available to clinicians in real time [25]. All other biomarkers were measured in batches after clinical data collection and follow up had been complete. Results were not available to clinicians. Other than CRP, biomarker results were not used by the research team in SBI categorisation.

PCT was measured using an automated analyser (KRYPTOR-TRACE technology, BRAHMS, Germany). The aPTT biphasic waveform was measured using an MDA-180 analyser (Trinity Biotech, Ireland) [12]. CRP was measured using an immunoturbidimetric assay on an automated Abbott Architect analyser. NGAL (R&D systems, Minneapolis) and resistin (Assaypro, Missouri), were carried out on a random sample of patients and measured using commercially available ELISAs according to the manufacturer's instructions.

## Statistical analysis

Descriptive statistics were calculated to summarise clinical data available for study participants with and without SBI in terms of baseline patient characteristics, patient outcomes and baseline biomarker measurements. The following univariate tests were applied to identify which variables had a statistically significant association with SBI. For comparing continuous variables with SBI, a t-test was applied if the variable was normally distributed, otherwise a Mann Whitney U test was used. For comparing categorical variables with SBI, a Pearson's chi-squared test or Fisher's exact test was applied.

The primary analysis compared the performance of the different study biomarkers and combinations of biomarkers for diagnosis of SBI. Receiver operator curves (ROC) analysis was undertaken for individual and combinations of biomarkers using logistic regression including the biomarker as a continuous variable. Participants were only included in the model if biomarker data were available, and values were not imputed for the missing biomarker data. The optimal cut off point at which to diagnose an SBI was calculated for each ROC curve using the Youden Index [26] with the sensitivity, specificity, positive predictive value (PPV), negative predictive value (NPV) and positive/negative likelihood ratios reported. In addition, performance of biomarker combinations at various cut points, defined a priori, were calculated and performance calculated using logistic regression.

Net reclassification improvement (NRI) was used to measure the additive value of measuring multiple biomarkers and to quantify improvement in performance over CRP alone which is the most commonly used biomarker in UK paediatric clinical practice [26]. The magnitude of NRI is more important than the statistical significance, therefore we reported NRI with

confidence intervals, as well as significance values [26, 27]. To investigate the trend in bio-markers over the first 7 days of PICU admission for children with and without SBI, we fitted linear mixed effect models including an interaction effect for time and SBI status. A quadratic term was included in the model when the biomarker variation over time was nonlinear.

Statistical methods for secondary outcomes measures are provided in the Supporting Statistical Methods. Statistical Analyses were performed using R v3.4.0, with specific packages ROCR and Hmisc to estimate ROC curve and NRI, respectively. Statistical methods for secondary outcomes measures are provided in the Supporting Statistical Methods.

## Results

### Participants

A total of 2468 children were admitted to the PICU from October 2010 to June 2012; 1339 were not eligible for inclusion and consent could not be obtained in 472 cases, therefore 657 patients were enrolled (Fig 1). The median age of children included was 1.01 years (IQR 0.30–4.99 years). Baseline characteristics of participants are shown in S2 Table in S1 File. Most patients (473/657, 72.0%) were admitted to PICU post-operatively, most frequently after cardiac surgery (368/657, 56.0%).

### Serious bacterial infection on admission

A total of 92 participants (14.0%) fulfilled study criteria for SBI on admission to the PICU. Consensus for presence of SBI according to study criteria was met in the multi-disciplinary PICU infection meeting in 90 cases. Two cases, in whom consensus was not met in the weekly meeting, were retrospectively reviewed in detail by the study experts and classified as meeting SBI criteria. Seventy -seven SBI's were community acquired and 15 were hospital-acquired. Twenty-one infections represented bacterial-viral coinfections, 65 were confirmed bacterial infection and 6 were presumed bacterial infections. Sterile site bacterial culture results and the clinical SBI diagnosis for these patients are shown in S3 and S4 Tables in S1 File. The six patients with presumed serious bacterial infection included two cases of necrotising enterocolitis, peritonitis post bladder rupture, clinical sepsis secondary to round pneumonia on chest x ray and septic shock secondary to sinusitis which was confirmed on CT. Thirty patients fulfilled consensus definitions [28] for sepsis, 26 for septic shock, 18 for severe sepsis and 13 for infection on admission.

Outcomes of patients with and without SBI on admission are compared in Table 1. Twenty-eight-day mortality (8.70% vs 1.59%, (p<0.001)), median duration of PICU stay (5.4 days vs 2.9 days, p<0.001) and duration of invasive ventilation (3.8 days' vs 1.7 days, p < 0.001) were greater in the SBI group.

### Diagnostic performance of biomarker results

There was a significant difference in median baseline biomarker measurement between patients with SBI at admission and those without (S5 Table in S1 File). Sensitivity, specificity, PPV and NPV, LR and cut points for individual biomarkers and combinations are show in Table 2.

Comparing individual biomarkers using ROC curves, diagnosis of SBI was best with CRP (AUC 89.73, 95% CI 85.83 to 93.69), followed by PCT (AUC 83.73 95% CI 78.11 to 89.35). Combinations of biomarkers were superior to individual biomarkers alone (Fig 2). Performance of combinations of biomarkers based upon specific cut points are shown in S5 Table in

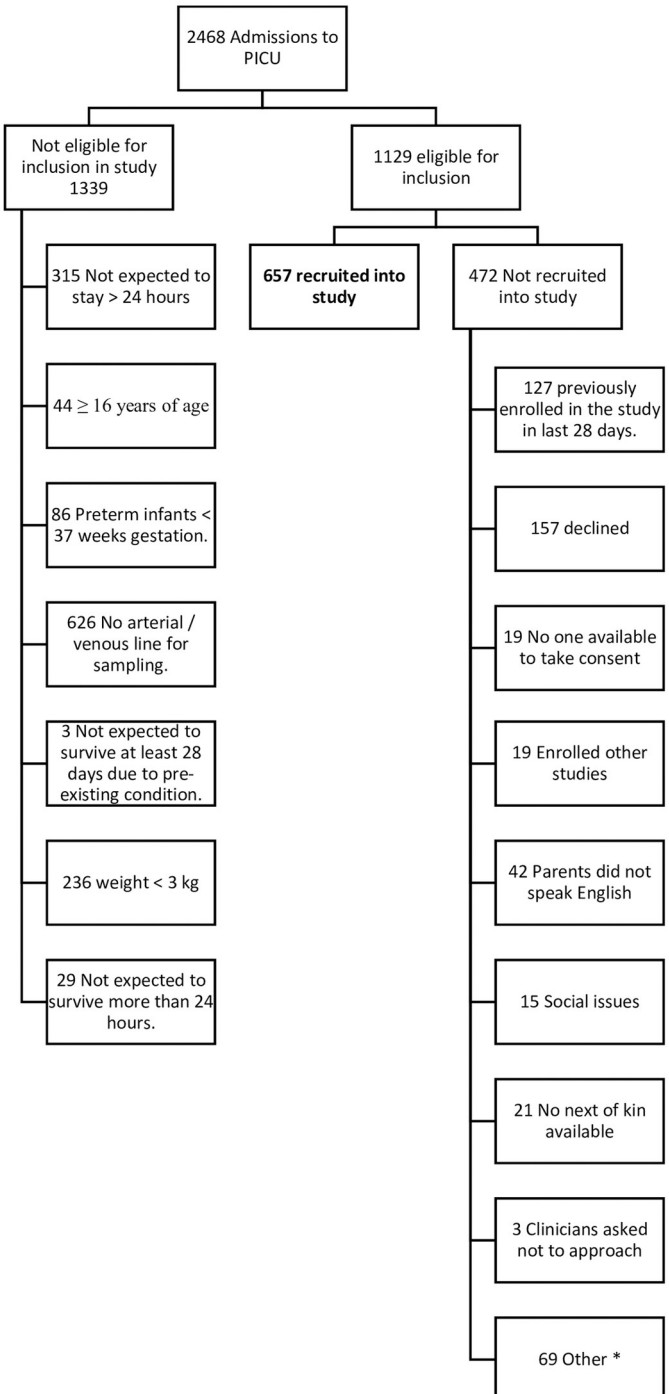

*Parents unable to give consent, previously declined to participate, previously enrolled in the study during the same illness episode and now ≥28 days.

**Fig 1. Patient flow diagram.** *Parents unable to give consent, previously declined to participate, previously enrolled in the study during the same illness episode and now ≥28 days.

**Table 1. Comparison of patient outcome measures for patients with and without SBI on admission.**

| | SBI on Admission | | Total | p-value |
|---|---|---|---|---|
| | No | Yes | | |
| **Patient Outcome** | *N = 565 (86%)* | *N = 92 (14%)* | *N = 657* | |
| **DIC, n (%)** | | | | |
| No | 502 | 86 | 588 | 0.246 |
| Yes | 63 (11) | 6 (7) | 69 (11) | |
| **MODS, n (%)** | | | | |
| No | 485 | 48 | 533 | <0.001 |
| Yes | 80 (14) | 44 (48) | 124 (19) | |
| **28 Day Mortality n (%)** | | | | |
| Alive | 556 | 84 | 640 | <0.001 |
| Dead | 9(2) | 8(9) | 17(3) | |
| **Duration of PICU stay (days)** | | | | |
| Mean (SD); range | 6.4(10.3);0.3–127.1 | 8.2 (9.3); 0.4–46.0 | 6.6 (10.2);0.3–127.1 | <0.001 |
| Median (IQR) | 2.9 (1.7–6.6) | 5.4 (3.1–8.4) | 3.3 (2.7) | |
| **Renal Replacement Therapy, n (%)** | | | | |
| No | 524 | 86 | 610 | 0.972 |
| Yes | 41 (7.3) | 6 (6.5) | 47 (7.2) | |
| **Maximum PELOD** | N = 563 | N = 91 | N = 654 | |
| Mean (SD); range | 12.2 (6.4); 0.0–43.0 | 13.0 (5.2); 1.0–32.0 | 12.1 (6.3); 0.0–43.0 | 0.439 |
| Median IQR | 12.0 (11.0–12.0) | 12.0 (11.0–13.0) | 12.0 (11.0–12.0) | |
| **Maximum temperature in 1ˢᵗ 48 hours** | N = 538 | N = 90 | N = 628 | |
| Mean (SD); range | 37.3 (0.8); 33.7–39.5 | 37.6 (1.0); 34.0–41.3 | 37.3 (0.8);33.7–41.3 | 0.002 |
| Median (IQR) | 37.2 (36.8–37.8) | 37.5 (37.1–38.0) | 37.3 (36.8–37.8) | |
| **Duration of IPPV** | N = 524 | N = 77 | N = 601 | |
| Mean (SD); range | 3.7 (5.7); 0.1–66.1 | 3.8 (5.6); 0.4–35.5 | 3.9 (5.7); 0.1–66.1 | <0.001 |
| Median (IQR) | 1.7 (0.5–4.4) | 3.8 (2.0–5.9) | 1.9 (0.6–5.1) | |
| **Number in whom antibiotics given** | 567 | 92 | 639 | |
| Duration of antibiotics | N = 499 | N = 81 | N = 580 | |
| Mean (SD); range | 4.6 (4.3); 1.0–29.0 | 8.9 (8.8); 1.0–70.0 | 5.3 (5.4); 1.0–70.0 | <0.001 |
| Median (IQR) | 3.0 (2.0–6.0) | 7.0 (5.0–10.0) | 3.0 (2.0–7.0) | |

S1 File. Maximum specificity (76.1%) was achieved when considering PCT > 1.25 and CRP > 25, whilst maximum sensitivity (97.6%) was found with a six-biomarker combination.

Calculating NRI (Table 3) for combinations of biomarkers, all five models significantly increased the proportion of patients correctly classified as not having an SBI. Two combinations, Model 3 (PCT, resistin and CRP) and Model 4 (PCT with CRP) significantly reduced the number of patients correctly classified as having the event. The combination of PCT, resistin, plasma NGAL with CRP resulted in the greatest net gain in classification proportion of 0.69 (p<0.001, 95% CI 0.33 to 1.04) with 10.5% reduction in correct classification of patients with SBI (p = 0.52, 95% CI -0.42 to 0.21) but a 79% improvement in correct classification of patients without events (p<0.001 95% CI 0.63 to 0.95).

## Longitudinal biomarker measurements

We plotted longitudinal biomarker profiles for median and mean NGAL, resistin, PCT and CRP. Only PCT and CRP demonstrated clearly distinct profiles between children with SBI and without SBI (Fig 3). Linear mixed effect models were fitted for longitudinally repeated data of

**Table 2. Accuracy and cut points of individual biomarkers and combinations of biomarkers for diagnosis of SBI on admission to PICU.**

| Individual biomarker or biomarker combinations AUC (95% CI) | Cut-point# | Sensitivity 95% CIs | Specificity 95% CIs | PPV 95% CIs | NPV 95% CIs | LR+ 95% CIs | LR- 95% CIs |
|---|---|---|---|---|---|---|---|
| PCT (ng/ml) * 83.73 (78.11, 89.35) | 1.25 | 0.78 (0.69, 0.88) | 0.80 (0.76, 0.84) | 0.37 (0.26, 0.48) | 0.96 (0.94, 0.98) | 3.92 (3.17, 4.85) | 0.27 (0.17, 0.42) |
| Urine NGAL (ng/ml) 56.72 (48.18, 65.25) | 95.10 | 0.47 (0.34, 0.60) | 0.67 (0.62, 0.71) | 0.17 (0.07, 0.27) | 0.90 (0.87, 0.93) | 1.41 (1.03, 1.93) | 0.79 (0.61, 1.03) |
| Plasma NGAL (ng/ml) 68.07 (60.09, 76.06) | 194.00 | 0.61 (0.49, 0.73) | 0.75 (0.69, 0.81) | 0.38 (0.26, 0.50) | 0.88 (0.84, 0.93) | 2.44 (1.81, 3.30) | 0.52 (0.37, 0.72) |
| Resistin (ng/ml) 80.17 (73.14, 87.20) | 96.28 | 0.64 (0.51, 0.76) | 0.85 (0.80, 0.90) | 0.51 (0.38, 0.65) | 0.90 (0.86, 0.94) | 4.20 (2.90, 6.10) | 0.43 (0.30, 0.61) |
| APTT slope 65.90 (58.89, 72.92) | -0.03 | 0.46 (0.35, 0.57) | 0.84 (0.81, 0.87) | 0.33 (0.23, 0.43) | 0.90 (0.87, 0.93) | 2.86 (2.09, 3.92) | 0.65 (0.53, 0.79) |
| APTT TR18 (%) 67.23 (60.29, 74.17) | 99.40 | 0.51 (0.40, 0.62) | 0.84 (0.81, 0.87) | 0.35 (0.25, 0.45) | 0.91 (0.88, 0.93) | 3.13 (2.32, 4.22) | 0.59 (0.47, 0.74) |
| CRP mg/l 89.76 (85.83, 93.69) | 2.09 | 0.93 (0.87, 0.99) | 0.76 (0.72, 0.79) | 0.37 (0.25, 0.48) | 0.99 (0.98, 1.00) | 3.81 (3.21, 4.52) | 0.09 (0.04, 0.22) |
| Model (1) ‡ PCT, Resistin, pNGAL, CRP 91.64 (87.58, 95.70) | 0.12 | 0.92 (0.84, 1.00) | 0.83 (0.77, 0.89) | 0.57 (0.42, 0.73) | 0.98 (0.95, 1.00) | 5.42 (3.77, 7.79) | 0.10 (0.03, 0.28) |
| Model (2) ‡ PCT, Resistin, APTT TR18, CRP 92.52 (88.22, 96.81) | 0.08 | 0.97 (0.92, 1.00) | 0.75 (0.67, 0.82) | 0.52 (0.35, 0.68) | 0.99 (0.97, 1.00) | 3.83 (2.82, 5.19) | 0.04 (0.01, 0.27) |
| Model (3) ‡ PCT, Resistin, CRP 92.36 (88.54, 96.17) | 0.10 | 0.95 (0.88, 1.00) | 0.79 (0.73, 0.86) | 0.54 (0.38, 0.69) | 0.98 (0.96, 1.00) | 4.54 (3.30, 6.23) | 0.06 (0.02, 0.25) |
| Model (4) ‡ PCT, CRP 92.66 (88.47, 96.85) | 0.06 | 0.92 (0.85, 0.99) | 0.86 (0.83 0.90) | 0.50 (0.37, 0.62) | 0.99 (0.97, 1.00) | 6.67 (5.18, 8.58) | 0.10 (0.04, 0.22) |

*Pre-test probability is 12.89%.

#Cut-points were determined using the Youden Index, and *probability* cut-points were given for biomarker combinations from models (1)-(4), see below.

‡Prediction via logistic regression models:.

Model (1): $p = -2.68 + 6.31 \times 10^{-3} \; PCT + 1.45 \times 10^{-3} \; Resistin + 0.46 \times 10^{-3} \; pNGAL + 14.79 \times 10^{-3} \; CRP$.

Model (2): $p = 0.85 + 5.64 \times 10^{-3} \; PCT + 1.73 \times 10^{-3} Resistin - 36.34 \times 10^{-3} \; APTT + 20.10 \times 10^{-3} \; CRP$.

Model (3): $p = -2.60 + 7.29 \times 10^{-3} \; PCT + 1.73 \times 10^{-3} \; Resistin + 15.16 \times 10^{-3} \; CRP$.

Model (4): $p = -2.96 + 27.02 \times 10^{-3} \; PCT + 18.49 \times 10^{-3} \; CRP$.

the 4 biomarkers. The difference between profiles for those with and without SBI was determined by including an interaction term for time and SBI status. A quadratic term was included in the model for CRP as the variation over time was highly nonlinear. The corresponding p values for the interaction effect were 0.116, 0.000, 0.230 and 0.000 for plasma NGAL, PCT, Resistin and CRP respectively, supporting the observation in Fig 3 that only PCT and CRP demonstrated clearly distinct longitudinal profiles between children with and without SBI.

## Prognostic performance of biomarkers

**28 day mortality.** A total of 640 patients (97.41%) were alive at 28 days. Independent variables identified as being individually associated with 28-day mortality (at the 5% significance level) were surgery (none, non-cardiac), unplanned admission, SBI, fever, recent antibiotics, immunosuppression, PCT and admission PELOD. The number of deaths (n = 17) was insufficient to fit a statistical model.

**Prolonged ICU stay and duration of ICU stay.** The median duration of PICU stay was 3.3 days (IQR 2.0–7.0 days). The independent variables identified as being significantly associated with prolonged ICU stay are shown in S7 Table in S1 File. A stepwise regression model to predict prolonged PICU stay demonstrated an AUC of 80.2%, and included the following variables: type of admission, inotrope score, age, recruitment to the CHIP study, long term

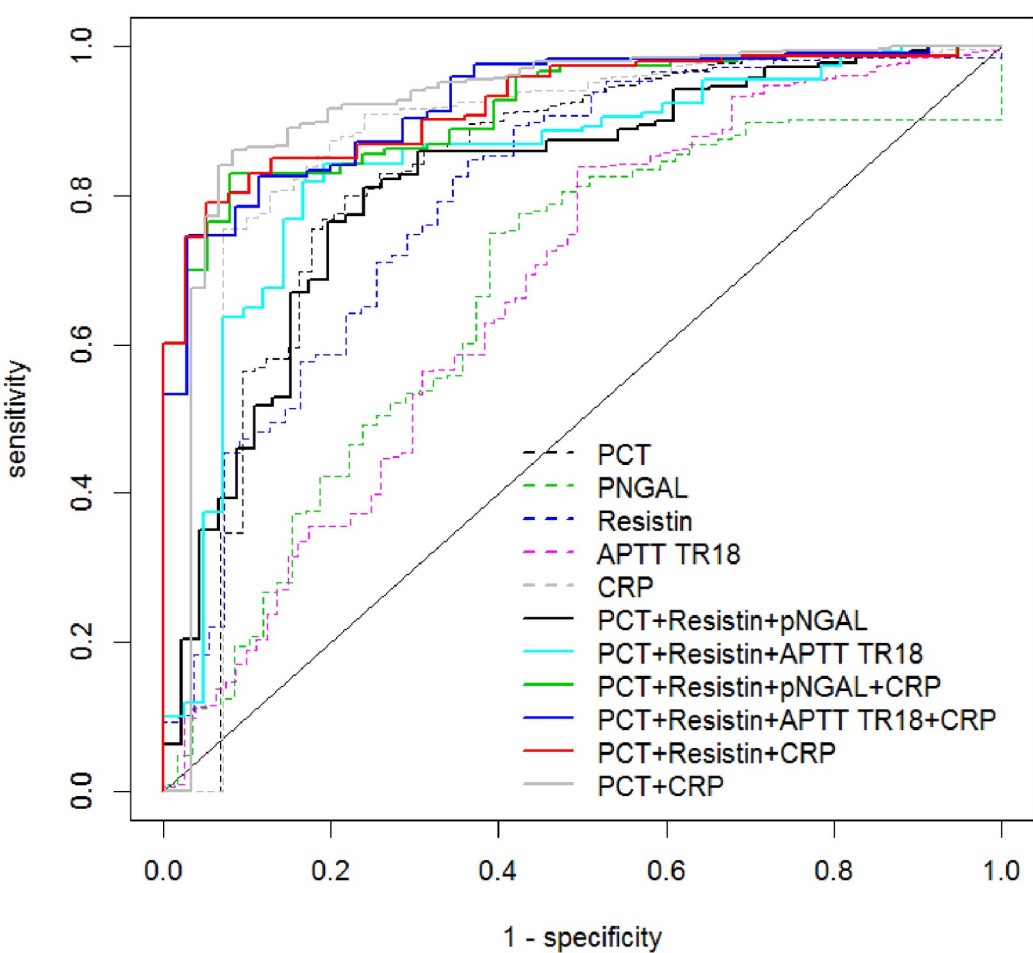

| Biomarker | AUC (95% CI) |
|---|---|
| PCT | 83.73 (78.11, 89.35) |
| Resistin | 80.17 (73.14, 87.20) |
| CRP | 89.76 (85.83, 93.69) |
| Plasma NGAL | 68.07 (60.09, 76.06) |
| APTT TR18 | 67.23 (60.29, 74.17) |
| PCT+ Resistin+ Plasma NGAL | 81.85 (74.81, 88.88) |
| PCT+ Resistin+ APTT TR18 | 84.57 (77.58, 91.55) |
| PCT+Resistin+CRP+ Plasma NGAL | 91.64 (87.58, 95.70) |
| PCT+Resistin+CRP+ APTT TR18 | 92.52 (88.22, 96.81) |
| PCT+Resistin+CRP | 92.36 (88.54, 96.17) |
| PCT+CRP | 92.66 (88.47, 96.85) |
| PCT+Resistin | 84.38 (77.70, 91.07) |
| Resistin+CRP | 92.31 (88.77, 95.85) |

**Fig 2. Receiver operating characteristic curve for individual and combinations of biomarkers for detection of SBI on admission to PICU.**

antibiotics, antibiotics prior to ward admission, PELOD, platelet count, lactate count. Patients with unplanned admissions, through the emergency department or following surgery, had greater odds of prolonged PICU stay compared to planned admissions, OR 5.6 (95% CI 3.20 to 10.19) and 8.3 (95% CI 2.32 to 34.11) respectively). Long-term antibiotics were associated with

**Table 3.** Net reclassification improvement (NRI) in new prediction models (biomarker combinations) versus current C-reactive protein alone model.

| New model | Summary NRI (95% CI), p-value | NRI for events (95% CI), p-value | NRI for non-events (95% CI), p-value |
|---|---|---|---|
| Model (1) ‡ PCT, Resistin, pNGAL, CRP | 0.686 (0.330, 1.041), <0.001 | -0.105 (-0.423, 0.213), 0.516 | 0.791 (0.632, 0.949), <0.001 |
| Model (2) ‡ PCT, Resistin, APTT TR18, CRP | 0.616 (0.241, 0.990), 0.001 | -0.257 (-0.588, 0.074), 0.128 | 0.873 (0.698, 1.048), <0.001 |
| Model (3) ‡ PCT, Resistin, CRP | 0.524 (0.172, 0.876), 0.004 | -0.385 (-0.698, -0.071), 0.016 | 0.908 (0.750, 1.067), <0.001 |
| Model (4) ‡ PCT, CRP | 0.610 (0.341, 0.879), <0.001 | -0.279 (-0.530, -0.0277), 0.030 | 0.889 (0.793, 0.985), < 0.001 |
| Model (5) Resistin + CRP | 0.715 (0.385, 1.045), <0.001 | -0.227 (-0.523, 0.068), 0.132 | 0.943 (0.794, 1.091), <0.001 |

increased odds of prolonged PICU stay (4.9 OR, 95% CI 1.66 to 16.98, p = 0.005) whereas antibiotics prior to admission reduced the odds (0.5 OR, 95% CI 0.28–0.99, p = 0.049) (S8 Table and S1 Fig in S1 File).

The independent variables significantly associated with length of ICU stay (at the 5% significance level) are shown in S9 Table in S1 File. Four hundred and forty-seven patients had a full record of observations and so were included in the model. The model of best fit for log (length of PICU) is shown in S10 Table in S1 File. Variables in the model include age, surgery, type of admission, bypass time, admission to CHiP study, inotrope score, long term antibiotics, recent antibiotics, antibiotics prior to ward, general appearance, PCT and PELOD. For example, we find that each 1-year increase in age multiplies the expected value of length of stay by 0.96 (95% CI of 0.94 to 0.98, p <0.01). The $R^2$ value is 25.9% which suggests that the predictive capacity of this model is low.

**Duration of IPPV.** The median duration of IPPV was 1.9 days (IQR 0.6 to 5.1). A total of 326 out of 657 patients had no missing observations and were included in the model. The independent variables significantly associated with duration of ventilation are shown in S11 Table in S1 File. The model of best fit for log (duration of IPPV) is shown in S12 Table in S1 File. Increased duration of invasive ventilation was associated with decreasing age, increasing bypass time, unplanned admissions, no or non-cardiac surgery, PELOD, BE, lactate and inotrope scores.

To further investigate the prognostic values of the biomarkers, first recorded and maximum blood lymphocyte, neutrophil and platelet count, plasma NGAL, CRP and PCT were analysed as predictors of prolonged PICU stay using a logistic regression model. The model of best fit for the initial values included the independent variables lymphocyte and PCT whilst the model for the maximum values included lymphocyte, CRP, platelet and plasma NGAL (Table 4). For each model, we determined the corresponding ROC curves (S2 & S3 Figs in S1 File) and used these curves to identify the optimal cut-off at which to classify a patient has a prolonged PICU stay. The predictive performance of the model including the maximum biomarker values was higher predictive performance than the first recorded biomarker model (AUC 78.95%, 95% CI 73.73% to 84.16% v AUC 68.05%, 95% CI 63.93% to 72.16%).

## Discussion

This is the largest and most comprehensive study to date investigating the performance of multiple biomarkers for the diagnosis of SBI in all children admitted to PICU. We demonstrate improved diagnostic performance using combinations of biomarkers compared to CRP alone.

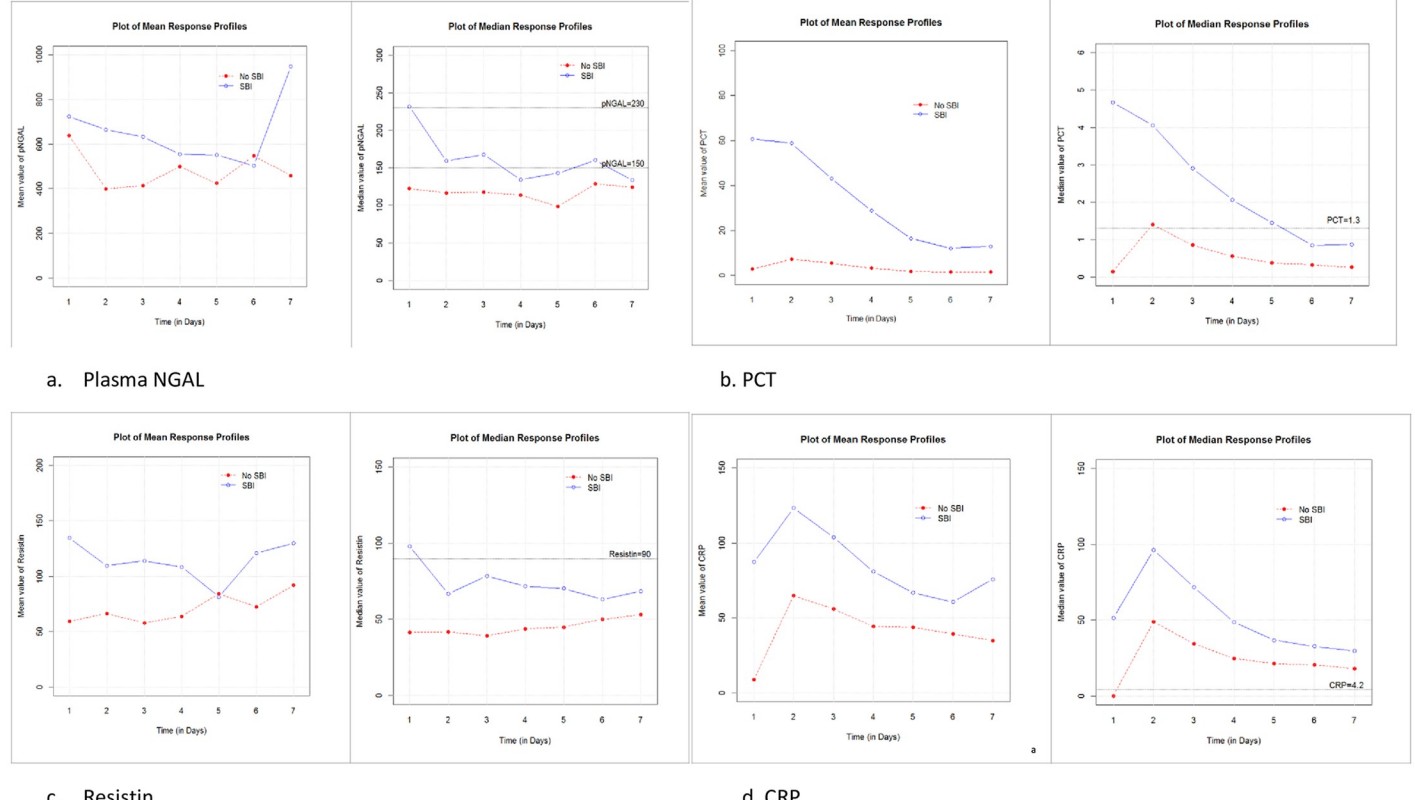

**Fig 3. Longitudinal trends in mean and median biomarkers over first seven days of PICU admission.** a. Plasma NGAL; b. PCT; c. Resistin; d. CRP.

Fourteen percent of admissions were diagnosed with SBI which is comparable to data from other tertiary PICUs [1, 29, 30]. Overall the prevalence of underlying chronic disease of 86.4% in the cohort may be indicative of the changing epidemiology of critically ill children [31]. Children with SBI on admission had significantly increased 28-day mortality, higher rates of MODS and longer ICU stays compared to those without SBI, highlighting the significant burden of sepsis despite widespread use of bacterial conjugate vaccines.

Individually, CRP was the most sensitive biomarker for diagnosis of SBI on PICU admission whilst resistin was the most specific. In our cohort SBI patients were predominantly referrals from other hospitals, which is representative of the structure of PICU in the U.K. We

**Table 4. Final multivariable model of best fit for prolonged PICU stay using the first recorded or maximum values of the blood biomarkers.**

| First Recorded Values | Coefficient | Standard Error | Odds Ratio | OR 95% CI | P value |
|---|---|---|---|---|---|
| Intercept | 0.139 | 0.114 | 1.149 | 0.9198–1.4401 | 0.224 |
| Log2(lymphocyte) | 0.082 | 0.080 | 1.085 | 0.9281–1.2694 | 0.306 |
| Log2(PCT) | 0.213 | 0.029 | 1.238 | 1.1704–1.3128 | <0.001 |
| **Maximum Value** | | | | | |
| Intercept | -10.461 | 2.023 | 0 | (0–0.0013) | 0 |
| Log2(lymphocytes) | 0.750 | 0.169 | 2.118 | (1.5375–2.9861) | 0 |
| Log2(platelet) | 0.743 | 0.209 | 2.102 | (1.4059–3.2015) | <0.001 |
| Log2(plasma NGAL) | 0.215 | 0.094 | 1.240 | (1.0323–1.4939) | 0.022 |
| Log2(CRP) | 0.323 | 0.140 | 1.380 | (1.051–1.8288) | 0.022 |

studied biomarkers on admission to PICU rather than on initial presentation. This may account for the superior performance of CRP compared to PCT which is generally thought to show a slower response to infection than PCT. All combinations of biomarkers, other than PCT and resistin, showed excellent discriminatory ability with AUC > 90%. PCT with CRP was best for ruling in SBI, with a positive likelihood ratio of 6.67 (95% CI 5.18 to 8.38). To rule out SBI, the combination ofPCT, resistin, aPTT TR18 and CRP showed the lowest likelihood ratio at 0.04 (95% CI 0.04 to 0.010). Overall, based upon net reclassification improvement the combination of PCT, resistin, pNGAL and CRP yielded the most improved correct classification of events.

Secondary analysis investigating variables predictive of predictive of duration of ventilation, duration of and prolonged PICU stay did not find biomarkers to be significant predictors of these adverse outcomes. Maximum values for lymphocytes, NGAL, CRP and platelets as opposed to biomarker values on admission, provided a better model to predict the risk of prolonged PICU stay, suggesting that these biomarkers have prognostic value. Risk stratification of patients with suspected sepsis, based on biomarkers and clinical indices, has been recommended as an area for future research [1] (1). Longitudinal profiles for PCT showed the greatest percentage drop in values over the first seven days of therapy in children with SBI, suggesting that PCT might be useful in guiding duration of antimicrobial therapy in children, as has been shown in neonates and adults [32, 33]. Multi-centre randomised controlled trials aiming to determine if the addition of PCT testing to current best practice can safely allow a reduction in duration of antimicrobial therapy in hospitalised children with suspected or confirmed bacterial infection compared to current best practice [34].

Younger age was found to be a predictor of longer duration of PICU stay and IPPV. There was a higher risk of prolonged PICU stay and IPPV associated with unplanned versus planned admissions. Non-cardiac surgical admission was also a risk factor for adverse outcomes. The odds of prolonged PICU stay was almost five times greater for a patient who received long term antibiotics compared with a patient who did not. This may reflect the development of antimicrobial resistance or that patients with more comorbidities maybe more likely to be on long term antibiotics prophylaxis.

## Strengths and limitations

The current study included 657 patients with 92 cases (14%) of SBI on admission, and therefore the study is underpowered to achieve 80% sensitivity and specificity for a single biomarker. However, to the best of our knowledge, this is the largest study to prospectively assess the performance of multiple biomarkers of SBI in a heterogeneous cohort of critically ill children and uniquely profiles longitudinal biomarker changes within the cohort. We studied over 600 children and collected comprehensive clinical and laboratory data with follow up to 28 days after admission, which makes our findings generalizable to other tertiary paediatric settings. The enrolment of all consecutive admissions to PICU, as opposed to just those with suspected sepsis or a positive microbiology result, overcomes spectrum bias where a biomarker performs better in those more likely to have the disease. A limitation of the study was that, as this was a pragmatic cohort study, there were large numbers of children admitted postoperatively without infection, which may have underestimated biomarker performance, but we aimed to replicate real-life prediction of events at admission. A significant proportion of included patients were transfers from other hospital; hence timing of biomarker measurement in relation to disease onset was variable in the cohort. The high number of missing values for resistin and pNGAL is a further limitation.

## Conclusion

In this study of unselected, critically ill children admitted to PICU, combinations of biomarkers including of PCT and CRP provided excellent discrimination of SBI on admission. SBI on admission was associated with increased 28-day mortality and longer duration PICU stay. Early recognition of sepsis should lead to better outcomes from children admitted to PICU.

## Supporting information

**S1 File.**
(DOCX)

## Acknowledgments

The authors would like to thank the children, young people, parents and carers who participated in the study. We also thank the PICU staff who contributed to the study.

## Author Contributions

**Conceptualization:** Stephane Paulus, Enitan D. Carrol.

**Data curation:** Maryke J. Nielsen, Enitan D. Carrol.

**Formal analysis:** Maryke J. Nielsen, Ruwanthi Kolamunnage-Dona, Christine Chesters, Graham Jeffers, Colin Downey, Lynsey McColl, Anthony McKeever.

**Funding acquisition:** Enitan D. Carrol.

**Investigation:** Paul Baines, Christine Chesters, Graham Jeffers, Caroline Broughton, Nigel Cunliffe, Enitan D. Carrol.

**Methodology:** Paul Baines, Rebecca Jennings, Sarah Siner, Ruwanthi Kolamunnage-Dona, Paul Newland, Enitan D. Carrol.

**Project administration:** Rebecca Jennings, Sarah Siner, Paul Newland, Matthew Peak, Jennifer Preston, Stephane Paulus, Nigel Cunliffe, Enitan D. Carrol.

**Resources:** Sarah Siner, Enitan D. Carrol.

**Supervision:** Paul Baines, Enitan D. Carrol.

**Writing – original draft:** Maryke J. Nielsen.

**Writing – review & editing:** Maryke J. Nielsen, Paul Baines, Rebecca Jennings, Sarah Siner, Ruwanthi Kolamunnage-Dona, Paul Newland, Matthew Peak, Christine Chesters, Graham Jeffers, Colin Downey, Caroline Broughton, Lynsey McColl, Jennifer Preston, Anthony McKeever, Stephane Paulus, Nigel Cunliffe, Enitan D. Carrol.

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
