## [Decision Letter · Decision Letter 0]

26 Aug 2020

PONE-D-20-03003

Procalcitonin, C-Reactive Protein, Neutrophil Gelatinase-Associated Lipocalin, Resistin and the APTT Waveform for the early diagnosis of serious bacterial infection and prediction of outcome in critically ill children.

PLOS ONE

Dear Dr. Nielsen,

Your submission has now been peer reviewed.  I agree that the manuscript would benefit from being revised according to the suggestions following and encourage you to do so.

Editor Comments to the Authors:

Please see the reviewer's comments. 

We look forward to receiving your revised manuscript.

Kind regards,

José Moreira, MD, MSc

Academic Editor

PLOS ONE

Journal Requirements:

2.Thank you for stating the following in the Financial Disclosure section:

[The study was funded jointly by the NIHR Liverpool Biomedical Research Centre in Microbial Diseases and the Alder Hey Charity awarded to EC.  MJN is supported by a Wellcome Trust Research Training Fellowship (award reference 203919/Z/16/Z).  The funders had no role in the study design, data collection and analysis, decision to publics, or preparation of the manuscript.  ].   

We note that one or more of the authors are employed by a commercial company: Select Statistical Services

Reviewers' comments:

Reviewer's Responses to Questions

**Comments to the Author**

1. Is the manuscript technically sound, and do the data support the conclusions?

Reviewer #1: Yes

Reviewer #2: Partly

2. Has the statistical analysis been performed appropriately and rigorously? 

Reviewer #1: Yes

Reviewer #2: Yes

3. Have the authors made all data underlying the findings in their manuscript fully available?

Reviewer #1: Yes

Reviewer #2: Yes

4. Is the manuscript presented in an intelligible fashion and written in standard English?

Reviewer #1: Yes

Reviewer #2: Yes

5. Review Comments to the Author

Reviewer #1: The study by Nielsen et al. is a prospective study in the UK evaluating the performance of an array of biomarkers on the diagnosis of SBI upon on admission to the PICU. The models including a combination of PCT, CRP and additional biomarkers indicated an improvement (as compared to CRP alone) in classification of SBI. An array biomarkers to identify SBI in pediatrics are intensely investigated and the literature is considerable in this area. The authors do a nice job on including different biomarkers in their analysis and showing improvement in the discrimination of SBI when using them in combination. Would suggest the following for this article:

1. Can you provide further detail on the diagnosis of SBI by your group specifically the level disagreement within the cases?

2. Are these commonly collected biomarkers for all ICU patients or specific to your study? Further are these POC biomarkers or are they commonly sent out to the laboratory. The external validity may be different if these are biomarkers are not commonly collected or available within real time.

3. I’m unsure whether providing the independent variables associated with prolonged ICU stay and duration of ICU stay is adding much the manuscript. The focus is on specific biomarkers and their combination. Including this analysis looks exploratory at best. Also, it would help the reader if you provide the 95% CIs along with the corresponding ORs, otherwise difficult to assess reliability of your point estimates.

4. It would be of more interest if we start to show the impact on clinical outcomes of these biomarkers and their combination rather than just their performance. Previous studies have evaluated biomarker performance for pediatrics and SBI with similar AUC findings.

Reviewer #2: A prospective observation study was conducted to investigate the accuracy of procalcitonin (PCT), neutrophil gelatinase-associated lipocalin (NGAL), resistin, activated partial thromboplastin time (aPTT) waveform and C-reactive protein (CRP) for their discriminative ability in diagnosing serious bacterial infections (SBI) in children. The combination of PCT, resistin, plasma NGAL and CRP resulted in the biggest net reclassification improvement compared to CRP alone.

Minor revisions:

1- Line 166: Identify the specific statistical univariate test applied to identify which variable had a statistically significant association with SBI.

2- Line 167: To improve clarity begin the sentence with: “For comparing continuous variables, …”

3- Line 168: To improve clarity begin the sentence with: “For comparing categorical variables, ...”

4- Line 169: Remove “if a count was below 5.” Fisher’s exact tests are appropriate to use when the expected, rather than observed, counts are less than 5.

5- Line 179: Provide a reference for net reclassification improvement.

6- Cite the statistical software used for the analysis.

7- Table 1 and Supplemental Table 5: Note what statistical methods were used to calculate the p-values. Explain that “Difference, (95% CI for location), p-value” pertains to continuous factors only. Possibly only the p-value is necessary.

8- Table 1: Small p-values are expressed as p< 0.001.

9- Line 170: Clarify the following statement. Possibly a more descriptive term can replace “association.” “The primary analysis examined the association between diagnosis of SBI and abnormal biomarkers.”

10- Line 219: Probabilities range from 0 to 1. It’s unclear what a probability of 12.89 represents.

11- P-values never equal zero; express small p-values as < 0.001.

12- Lines 255, 262, etc: Indicate the statistical methods used to identify independent variables.

13- Be sure to fully describe all statistical methods in the statistical methods section of the manuscript.

14- Table 4: Indicate if the models are univariate or multivariate.

15- Line 250: Provide a repeated measures analysis to support the conclusion that only PCT and CRP demonstrated clearly distinct profiles.

16- Supplementary Tables: Indicate the statistical method(s) used to obtain the best fitting models.

17- Add AUC values to the ROC curves.

18- State and justify the study’s target sample size with a pre-study statistical power calculation. The power calculation should include: sample size, alpha level (indicating one or two-sided), and statistical testing method.

6. PLOS authors have the option to publish the peer review history of their article (what does this mean?). If published, this will include your full peer review and any attached files.

Reviewer #1: No

Reviewer #2: No

---

## [Author Response · Author response to Decision Letter 0]

2 Dec 2020

Dear Editor 

Thank you for the reviewers’ comments on our Manuscript entitled “Procalcitonin, C-Reactive Protein, Neutrophil Gelatinase-Associated Lipocalin, Resistin and the APTT Waveform for the early diagnosis of serious bacterial infection and prediction of outcome in critically ill children”. We have considered the reviewers’ comments and have made the following amendments to the manuscript, figures and supporting information. 

• Funding Statement: We have amended the funding statement declaring the commercial affiliation of author LMcC and the role of her employer, Select Statistics, in the analysis of the study (Manuscript p2). We can confirm that Select Statistics provided support in the form of salaries for authors [LMcC], but did not have any additional role in the study design, data collection and analysis, decision to publish, or preparation of the manuscript. The specific roles of these authors are articulated in the ‘author contributions’ section.” 

• Competing Interests Statement: This has been updated on page 21 of the manuscript about author LMcC. “This does not alter our adherence to PLOS ONE policies on sharing data and materials”. 

Reviewer 1: 

1. Can you provide further detail on the diagnosis of SBI by your group specifically the level disagreement within the cases?

 Further details have been provided in relation to classification of patients with and without SBI by the multi-disciplinary PICU infection team meetings (Manuscript p12). 

2. Are these commonly collected biomarkers for all ICU patients or specific to your study? Further are these POC biomarkers or are they commonly sent out to the laboratory. The external validity may be different if these are biomarkers are not commonly collected or available within real time.

We can confirm that CRP was measured in the Alder Hey Hospital’s Clinical Chemistry Laboratory with results available to clinicians typically within hours of the sample being taken. As per standard UK clinical practice this was the only biomarker available to clinicians. All other biomarkers were measured in the laboratory after completion of clinical data collection and follow up. This has been clarified on page 8 and 9 of the manuscript. 

3. I’m unsure whether providing the independent variables associated with prolonged ICU stay and duration of ICU stay is adding much the manuscript. The focus is on specific biomarkers and their combination. Including this analysis looks exploratory at best. Also, it would help the reader if you provide the 95% CIs along with the corresponding ORs, otherwise difficult to assess reliability of your point estimates.

We have emphasized in the presentation of these results aimed to investigate the roles of the studied biomarkers in comparison to clinical variables as predictors of secondary outcome measures duration of ICU stay, prolonged ICU stay and duration of invasive positive pressure ventilation. 95% CI and significant levels have been added to all ORs specified in the manuscript. 

4. It would be of more interest if we start to show the impact on clinical outcomes of these biomarkers and their combination rather than just their performance. Previous studies have evaluated biomarker performance for pediatrics and SBI with similar AUC findings.

We note with thanks this comment strongly agree that it important that research can demonstrate the impact on clinical outcomes of these biomarkers. This project was a diagnostic accuracy study in which only CRP, a routinely used biomarker in the U.K., was available to clinicians and was therefore unable to demonstrate impact upon patient outcomes. This study has contributed to informing a multi-centre randomised control trial aiming to determine is the addition of procalcitonin testing to current best practice can safely allow a reduction in duration of antibiotic therapy in hospitalised children with suspected or confirmed bacterial infection compared to current best practice alone. https://www.isrctn.com/ISRCTN11369832?q=&filters=conditionCategory:Infections%20and%20Infestations,ageRange:Child&sort=&offset=7&totalResults=217&page=1&pageSize=10&searchType=basic-search. We have addressed this issue in the discussion (p19). 

Reviewer 2: 

1. Line 166: Identify the specific statistical univariate test applied to identify which variable had a statistically significant association with SBI – We have clarified the statistical tests used for this analysis (p9). 

2. Line 167: To improve clarity begin the sentence with: “For comparing continuous variables, …”. We have rephrased the sentences as per the reviewer’s advice. 

3. Line 168: To improve clarity begin the sentence with: “For comparing categorical variables, ...” We have rephrased the sentences as per the reviewer’s advice. 

4. Line 169: Remove “if a count was below 5.” Fisher’s exact tests are appropriate to use when the expected, rather than observed, counts are less than 5. We have removed “if a count was below 5” as per the reviewer’s recommendation. 

5. Line 179: Provide a reference for net reclassification improvement. Reference has been added (p10)

6. Cite the statistical software used for the analysis. Detailed have been added to statistical methods (p10)

7. Table 1 and Supplemental Table 5: Note what statistical methods were used to calculate the p-values. Explain that “Difference, (95% CI for location), p-value” pertains to continuous factors only. Possibly only the p-value is necessary. We concur with the reviewer’s comments and have removed the differences previously stated in Table 1. We have retained the difference results in Supplementary Table 5 since all variables are continuous variables. Statistical methods used to calculate these values have been added to the supporting statistical methods. 

8. Table 1: Small p-values are expressed as p< 0.001.- p values have been amended as recommended. 

9. Line 170: Clarify the following statement. Possibly a more descriptive term can replace “association.” “The primary analysis examined the association between diagnosis of SBI and abnormal biomarkers.” - This sentence has been replaced by the following “The primary analysis compared the performance of the different study biomarkers and combinations of biomarkers for diagnosis of SBI”. 

10. Line 219: Probabilities range from 0 to 1. It’s unclear what a probability of 12.89 represent - We have added in the previously missing % value to the pre-test probability and thank the reviewers for noting this omission. 

11. P-values never equal zero; express small p-values as < 0.001. – This has been amended throughout the manuscript. 

12. Lines 255, 262, etc: Indicate the statistical methods used to identify independent variables – This is included in supporting statistical methods. 

13. Be sure to fully describe all statistical methods in the statistical methods section of the manuscript – We have reviewed statistical methods in both the manuscript and supporting material and believe all methods are explained. 

14. Table 4: Indicate if the models are univariate or multivariate. – This has been added to the title of this table. 

15. Line 250: Provide a repeated measures analysis to support the conclusion that only PCT and CRP demonstrated clearly distinct profiles. – Details of this analysis and results are now included in the statistical methods (p10) and results section (p17). 

16. Supplementary Tables: Indicate the statistical method(s) used to obtain the best fitting models. - This is described in the supplementary statistical methods. 

17. Add AUC values to the ROC curves. - AUC values have been added to all ROC curves. 

18. State and justify the study’s target sample size with a pre-study statistical power calculation. The power calculation should include: sample size, alpha level (indicating one or two-sided), and statistical testing method.- We have included details of the apriori sample size calculation in the methods section of the manuscript and referred to this in the discussion. 

We thank the reviewers for their helpful comments, which have helped to improve the manuscript. We hope that all the comments have been satisfactorily addressed and that the manuscript will be deemed suitable for publication.

Yours sincerely 

Dr Maryke Nielsen 

Corresponding Author

---

## [Decision Letter · Decision Letter 1]

13 Jan 2021

Procalcitonin, C-Reactive Protein, Neutrophil Gelatinase-Associated Lipocalin, Resistin and the APTT Waveform for the early diagnosis of serious bacterial infection and prediction of outcome in critically ill children.

PONE-D-20-03003R1

Dear Dr. Nielsen,

We’re pleased to inform you that your manuscript has been judged scientifically suitable for publication and will be formally accepted for publication once it meets all outstanding technical requirements.

Kind regards,

Aleksandar R. Zivkovic

Academic Editor

PLOS ONE

---

## [Editor Report · Acceptance letter]

25 Jan 2021

PONE-D-20-03003R1 

Procalcitonin, C-Reactive Protein, Neutrophil Gelatinase-Associated Lipocalin, Resistin and the APTT Waveform for the early diagnosis of serious bacterial infection and prediction of outcome in critically ill children. 

Dear Dr. Nielsen:

I'm pleased to inform you that your manuscript has been deemed suitable for publication in PLOS ONE. Congratulations! Your manuscript is now with our production department. 

Kind regards, 

on behalf of

Dr. Aleksandar R. Zivkovic 

Academic Editor

PLOS ONE